# Transfer Learning in Trajectory Decoding: Sensor or Source Space?

**DOI:** 10.3390/s23073593

**Published:** 2023-03-30

**Authors:** Nitikorn Srisrisawang, Gernot R. Müller-Putz

**Affiliations:** 1Institute of Neural Engineering, Graz University of Technology, Stremayrgasse 16/IV, 8010 Graz, Austria; 2BioTechMed Graz, 8010 Graz, Austria

**Keywords:** electroencephalography, brain–computer interface, source localization, trajectory decoding, partial least-squares regression, unscented Kalman filter, transfer learning

## Abstract

In this study, across-participant and across-session transfer learning was investigated to minimize the calibration time of the brain–computer interface (BCI) system in the context of continuous hand trajectory decoding. We reanalyzed data from a study with 10 able-bodied participants across three sessions. A leave-one-participant-out (LOPO) model was utilized as a starting model. Recursive exponentially weighted partial least squares regression (REW-PLS) was employed to overcome the memory limitation due to the large pool of training data. We considered four scenarios: generalized with no update (Gen), generalized with cumulative update (GenC), and individual models with cumulative (IndC) and non-cumulative (Ind) updates, with each one trained with sensor-space features or source-space features. The decoding performance in generalized models (Gen and GenC) was lower than the chance level. In individual models, the cumulative update (IndC) showed no significant improvement over the non-cumulative model (Ind). The performance showed the decoder’s incapability to generalize across participants and sessions in this task. The results suggested that the best correlation could be achieved with the sensor-space individual model, despite additional anatomical information in the source-space features. The decoding pattern showed a more localized pattern around the precuneus over three sessions in Ind models.

## 1. Introduction

Since the discovery of the now well-known phenomenon called event-related desynchronization/synchronization (ERDS) [1] of electroencephalography (EEG), the so-called sensorimotor rhythm-based (SMR) brain–computer interface (BCI) [2] has been one of the main interests within the BCI field. Despite its promising accuracy, the SMR-based BCI is not very intuitive to control [3,4], as users need to map an execution (or imagery) of a motor function to a certain command, for example, eliciting foot movement imagery to send a command to move a cursor up, and this might not reflect how the brain functions. A more intuitive control scheme was proven possible in [5], and this opened another field of research focusing on inferring the target information (position/velocity of an on-screen target or an end-effector of a robotic arm) directly from a low-frequency EEG. Since then, there have been several studies that have tried to improve a decoder based on the so-called pursuit tracking tasks (PTT) [6,7,8,9,10,11,12,13] or center-out reaching tasks [14,15,16,17]. Most of them focused on improving the decoding performance; however, other aspects, such as minimizing the calibration time and mutual learning between the user and the machine [18], are usually not the main interest.

A typical machine learning model relies on the ideal assumption that the training and test datasets are sampled from the same distribution. However, this is not true in many cases, for example, due to a covariate shift [19] or in a case in which features are derived from EEG signals, due to their non-stationary properties [20,21]. The machine learning model must be trained again when this assumption is not fulfilled; otherwise, the model might struggle to perform. This process involves acquiring training data and computing a new model, which is expensive in terms of time and resources in some applications, such as in BCIs. So, transfer learning has been one of the major challenges within machine learning and data mining [22] as well as in BCIs because of the long time usually required to collect the data. This is because the participant must repeat the task several times so that the model can properly learn the intrinsic property of the EEG signal due to the characteristic low signal-to-noise (SNR) ratio in EEG. Despite the complex characteristics of the brain, transfer learning has been proven possible in several applications in BCI. However, most of the transfer learning research in BCIs has focused on SMR-based, P300-based, or passive BCIs [23,24], and to the best of our knowledge, no research has considered transfer learning in the context of continuous hand trajectory decoding.

Therefore, we wanted to investigate the feasibility of minimizing the time needed to collect training data via across-participant and across-session transfer learning and, as a result, maximizing the time of the actual usage of the BCI system. In the typical setting for BCI applications, brain signals corresponding to each participant performing a specific task are utilized to calibrate or “individualize” the model. To investigate such transfer learning, the model needs to learn general information for that task such that the model can be trained once and then transferred to another participant. It is then natural to investigate a larger dataset from several participants to find this task-specific common information. However, the memory needed to allocate the data may limit the size of the whole dataset. This depends on how the model is trained. For example, in an iterative approach such as the stochastic gradient descent (SGD), the dataset is divided into smaller batches, so the number of training samples is limited by the memory and not the size of the whole training sample. In contrast, for an approach such as the SIMPLS [25] algorithm used to train a partial least-squares (PLS) model, the whole dataset is represented in the memory during the training. Therefore, in this case, the number of training samples is limited by the amount of memory available in a computer or workstation. We applied recursive exponentially weighted PLS (REW-PLS) [26], which is based on a modified kernel algorithm in [27,28], to tackle the memory limitation. An extension of REW-PLS to a higher-dimensional tensor has already been applied in the field of BCIs [29,30].

Additionally, we wanted to investigate how different features extracted from EEG would generalize in these transfer learning scenarios. Two types of feature extraction were performed based on sensor-space EEG signals and source-space EEG signals, respectively. While the former involved processed EEG signals, the latter was performed via source localization of the processed EEG signals. Since the individual anatomical information from the head shape was utilized during the source localization process, we hypothesized that the model based on source-space features would represent a more common space across participants or sessions, leading to better generalization in the source-space model.

## 2. Materials and Methods

### 2.1. Dataset Description

The dataset considered in this study came from a study conducted across 3 sessions of an attempted movement with a 2D PTT by Pulferer et al. [10]. There were 10 able-bodied participants, each with 3 sessions of measurement. The mean age of the participants was 24 ± 5 years old. Written informed consent was obtained from the participants, who received monetary compensation for participating in the experiment. The experimental procedure was approved by the local ethics committee from the Medical University of Graz.

### 2.2. Experimental Paradigm

EEG and electrooculogram (EOG) signals were simultaneously acquired during the experiment. There were 60 channels of EEG and 4 channels of EOG acquired at 200 Hz. The task of the participant was to attempt to move their hand to follow a white target point on the screen (the snake) while their dominant hand movement was restricted. In the original study, there were 4 types of measurement blocks in each session: 0% EEG, 50% EEG, 100% EEG, and 100% EEG freerun, where the percentage indicates the mixing ratio of the decoded kinematics from EEG to the target snake (e.g., 0% EEG means 0% decoded kinematics and 100% target snake). No decoded kinematics were available in the 0% EEG block, so the delayed snake kinematics were shown instead. The feedback was given in the remaining measurement blocks by combining the decoded kinematics with the snake kinematics according to the specified percentages. There were two additional eye runs in between 0% EEG blocks in which the participant had to produce eye artifacts (horizontal and vertical eye saccades and blinking) according to a visual guide. The data from these eye runs were used to train artifact correction models (elaborated in the next section). The last block (100% EEG freerun) was excluded from this study. The experimental paradigm was similar to the previous study on PTT studies [6,7,8,9,13]. However, the participant controlled the cursor on the screen rather than a robotic arm.

### 2.3. Data Processing

The data processing pipeline was adapted from [12]. EEG and EOG were initially acquired at 200 Hz before being high-pass filtered at 0.18 Hz and then downsampled to 100 Hz. An eye artifact correction model, SGEYESUB [31], trained with the signals obtained during the eye runs, was applied to the EEG signals. The signals were re-referenced to the common average reference. A HEAR model [32], trained with eye-artifact-corrected signals from the eye runs, was applied to reduce the effects of pops and drifts from the signals. The signals were low-pass filtered at 3 Hz and then downsampled again to 20 Hz. The signals were expanded with their lags from −6 to 0 samples (corresponding to −300 to 0 ms). An additional step was applied only to the source-space decoding model: the representative signals were produced via source-space projection for the source-space model. The processing pipeline is summarized in Appendix A. The script to process the data was implemented in MATLAB R2019b (MathWorks Inc., Natick, MA, USA) via the EEGLAB toolbox [33].

### 2.4. Source Localization and Dimensionality Reduction

The measured EEG signals are typically modeled as mixed signals resulting from the underlying cortical sources or as the equation
(1)X=GJ+n
where $X\in R^{P\times T}$ is a matrix of measured EEG signals, $G\in R^{P\times Q}$ is a mixing or gain matrix, $J\in R^{Q\times T}$ is a matrix of underlying cortical sources, and $n\in R^{P\times T}$ is a matrix of additive noise. *P*, *Q*, and *T* represent the number of EEG channels, underlying sources, and time samples, respectively.

The processed EEG signals were first projected onto the source space. The forward and inverse problems were solved via Brainstorm [34]. The number of underlying sources was assumed to be 5000 evenly distributed cortical sources. Forward modeling was performed via a boundary element method (BEM) based on the ICBM152 template head model [35,36,37]. The relative conductivity of the three compartments (scalp, skull, and brain) was chosen as 0.41, 0.02, and 0.47, respectively, according to [38]. Electrode positions measured at the beginning of the experiment were utilized to co-register with the cortical surface from the ICBM152 template head model. These electrode positions represented the individual anatomical information of each participant in each session. There was no assumption about the cortical source orientation of cortical sources such that each cortical source was represented by 3 directional components (for x, y, and z directions in 3D), leading to a total of 15,000 source-space signals. The inverse model was solved via standardized Low-Resolution Electromagnetic Tomography (sLORETA) [39]. The noise covariance was estimated from the processed data from the eye runs after it was corrected with the SGEYESUB and HEAR artifact correction models.

The signals were then reduced by two folds. First, the source-space signals were restricted to only those within predefined regions of interest (ROIs). The ROIs were chosen according to [12] based on the MindBoggle atlas [40] around the central medial region of the brain, namely, the cuneus (CU), paracentral lobules (PCL), postcentral gyri (PoCG), precentral gyri (PreCG), precuneus (PCU), superior frontal gyri (SFG), occipital gyri (OG), and superior parietal lobules (SPL). The following ROIs were further divided into 2 smaller ROIs: SFG, PreCG, PoCG, SPL, PCU, and OG; this resulted in 28 ROIs, including both hemispheres. The location of ROIs is visualized in the results section. Then, representative signals were computed for each directional component of each ROI. The choice of a function to compute the representative signals was based on [12], which achieved the best performance by applying the principal component analysis (PCA) with 8 components, followed by averaging over the ROI, referred to as the Mean in that study. We chose to apply the Mean function, which exhibited slightly reduced performance but with a smaller number of source-space features (8 times smaller). Hence, there were 3 representative signals per ROI for 3 directional source components.

The projection of the measured sensor-space EEG signals and the dimensionality reduction can be summarized as
(2)$$y_{t}=UKx_{t}$$
where $y_{t}\in R^{3F\times 1}$ is a column vector representing the reduced source-space signals at time step *t*, $U\in R^{3F\times Q}$ is defined as the source-space scouting matrix, and $K\in R^{Q\times P}$ is a kernel matrix obtained by solving the inverse problem via sLORETA, which unmixes the assumed cortical sources from the measured EEG signals $x_{t}\in R^{P\times 1}$ at a time step *t*. *F* is the number of source-space features, where F=28 (1 representative signal × 28 ROIs). The element of the source-space scouting matrix, U, was defined as follows:(3)$$u_{ij}=\left\{\begin{array}{c} \frac{1}{R_{i}},\;j\in ROI_{i}\\ 0,\;otherwise \end{array}\right.$$
where $R_{i}$ represents the total number of sources within the *i*-th ROI, and *j* represents the index for the *j*-th source. This matrix summarizes the averaging over sources in each ROI.

### 2.5. Decoder

After obtaining the source-space signals, $y_{t}$, they were subjected to lag expansion, ${\tilde{y}}_{t}=\left[y_{t};y_{t-1};y_{t};\cdots ;y_{t-6}\right]$. The total number of features became 588 (3 directional source components × 28 ROIs × 1 representative signal × 7 time lags). This number was around 1.5 times higher than the number of features used in [10].

Simultaneously, the corresponding kinematics of the target snake, $z_{t}=\left[p_{hor,t};v_{hor,t};p_{ver,t};v_{ver,t}\right]$, were extracted, where $p_{i,t}$ and $v_{i,t}$ represent the position and velocity of dimension i (either horizontal, hor, or vertical, ver) at time t. The target snake kinematics were then extended to include non-directional information, namely, distance $d_{t}=\sqrt{p_{hor,t}{}^{2}+p_{ver,t}{}^{2}}$ and speed $s_{t}=\sqrt{v_{hor,t}{}^{2}+v_{ver,t}{}^{2}}$, and $\tilde{z_{t}}=ext\left(z_{t}\right)=\left[z_{t};d_{t};s_{t}\right]$. The pair of the extended source-space signals and the corresponding snake kinematics were utilized to train a cascaded model of recursive exponential weighted partial least-squares regression (REWPLS) [26] and a square-root unscented Kalman filter (SR-UKF) [41,42].

#### 2.5.1. Recursive Exponential Weighted PLS (REWPLS)

The training of a PLS model is typically performed via a built-in Matlab function called “PLSREGRESS”, which utilizes the SIMPLS algorithm [25]. However, this function does not work well in the case of large datasets due to memory constraints. One solution is to utilize a variation of PLS known as recursive exponential weighted PLS (REW-PLS) [26]. It was implemented via an improved kernel algorithm for PLS training [26,27], making use of a more compact representation of the data as covariance matrices rather than using the whole dataset. However, the computation of these covariance matrices still requires large memory, but this can be overcome by utilizing the recursive update of the covariance matrices. This recursive update scheme was intended to update the covariance matrices and the PLS model when new data are available, but we applied it to overcome the memory issue. The training dataset was divided into smaller batches. For each batch, the covariance matrices were recursively updated via these formulas:(4)$${\left(\tilde{Y}{\tilde{Y}}^{T}\right)}_{t}=\lambda {\left(\tilde{Y}{\tilde{Y}}^{T}\right)}_{t-1}+{\tilde{y}}_{t}{\tilde{y}}_{t}{}^{T}$$
(5)$${\left(\tilde{Y}{\tilde{Z}}^{T}\right)}_{t}=\lambda {\left(\tilde{Y}{\tilde{Z}}^{T}\right)}_{t-1}+{\tilde{y}}_{t}{\tilde{z_{t}}}^{T}$$
where $\tilde{Y}$ and $\tilde{Z}$ represent the predictor (in our case, the extended features) and response (in our case, extended snake kinematics) of time unit (samples or batches) t, and λ represents a forgetting factor, ranging between 0 and 1, that assigns weights between the covariance matrix of the previous time unit and the current time unit. In this case, we set λ=1 to set the weight equally for all time units. The term non-normalized covariance matrices might be more appropriate due to the lack of a normalization term, but these matrices will be referred to as covariance matrices for simplicity. Note that, as we set λ=1, the resulting (non-normalized) covariance matrices will be equivalent to computing the inner products from the whole dataset. These covariance matrices were used to train a kernel PLS model, which differs from the SIMPLS algorithm such that the inputs were the covariance matrices ($\tilde{Y}{\tilde{Y}}^{T},\tilde{Y}{\tilde{Z}}^{T}$) rather than the pair of source-space features and the intended snake kinematics ($\tilde{Y},\tilde{Z}$), as briefly discussed previously. The implementation of kernel PLS was from [43] based on modified kernel algorithm #2 [27]. One hyperparameter that needed to be optimized was the number of PLS latent components. The optimal hyperparameter was achieved by selecting the “knee” point on the curve in the plot of the mean Pearson’s correlations over the PLS-predicted kinematics from the 4 linear kinematics, $Z_{t}$, and the number of PLS latent spaces via 10-fold cross-validation (see Appendix A). The optimal number of PLS latent components was determined separately for each model.

#### 2.5.2. Square-Root Unscented Kalman Filter (SR-UKF)

The PLS latent space features, $\tilde{E}=W\tilde{Y}$, were then employed as an input pair ($\tilde{E},\tilde{Z}$) for the square-root unscented Kalman filter (SR-UKF), where W represents the PLS latent space projection matrix. The SR-UKF model was implemented according to [6,8]. The final outputs of the SR-UKF model were the predicted kinematics, $\hat{Z}$, which were employed together with the ground-truth snake kinematics to assess the model performance.

### 2.6. Simulated Inter-Session Transfer Learning Scenarios

The processed data were used to simulate an online experiment to assess the feasibility of performing an experiment where actual feedback is given from the beginning of the experiment. A leave-one-participant-out (LOPO) generalized model was trained on the data pool from the 0% EEG block of the first session of every participant except the data of the corresponding participant. This is illustrated in Figure 1, where the data of the respective participant (represented by red blocks in Figure 1) are not included in the training pool, and only the session 1 data for every participant (represented by green blocks in Figure 1) are included. The LOPO model was used as a starting model for each corresponding participant (see Figure 1; the Gen model was applied to 0% EEG of session 1 in all update strategies). The intensity of the red blocks indicates the corresponding session of the 0% EEG data. At the end of the 0% EEG measurement block, the decoding models were updated accordingly with the newly available data. The different update strategies are visualized in Figure 1. Four different update strategies (each with variations of sensor-space and source-space models) are described below.

#### 2.6.1. Generalized Model (Gen)

We examined how the generalized model performs in 3 sessions without introducing any of the data of the corresponding participant. The LOPO generalized model was applied throughout all the measurement blocks from the 3 sessions. The corresponding name of the model is simply Gen.

#### 2.6.2. Cumulative Generalized Model (GenC)

In this case, we examined how the generalized model performs given the cumulative updating of the data. The Gen model was applied to the 0% EEG block. The decoding model was cumulatively updated with the data from the 0% EEG block of session 1 from each respective participant, leading to the GenC1 model, which was applied to all of the measurement blocks until the 0% EEG block of session 2, where the model was cumulatively updated again with the new data (leading to the GenC2 model), and the same applied to session 3 as well (leading to the GenC3 model). This update strategy is visualized in Figure 1 by the additional red blocks, which represent the data of each respective participant from each session.

#### 2.6.3. Individual Model (Ind)

The Gen model was entirely replaced by the new decoding model trained with the newly available data, visualized as a single red block (Ind1 model in Figure 1). The decoding model was transferred to the 0% EEG block of the next session. Then, the decoding model was replaced by the decoding model trained with newly acquired data from the 0% EEG block of the current session and so on (leading to the Ind2 and Ind3 models, respectively). This is similar to how the model is typically utilized in a BCI study, where the data of the corresponding session are used only within the same sessions. Due to this similarity, the results could be roughly compared to those obtained in the original study [10].

#### 2.6.4. Cumulative Individual Model (IndC)

Similarly, the Gen model was used at the beginning of 0% EEG of session 1. Then, the Gen model was substituted by IndC1 (equivalent to Ind1 according to our naming convention). This model was trained with the data of all participants in session 1. This model was applied to the data until the end of 0% EEG of session 2, where the model was cumulatively updated again with the new data, leading to the IndC2 model. This also applied to session 3, leading to the IndC3 model at the end.

### 2.7. Performance Evaluation

#### 2.7.1. Decoding Performance

Two metrics were computed, Pearson’s correlation (r) and the signal-to-noise ratio (SNR), representing the similarity and dissimilarity between the ground-truth and decoded kinematic signals. The two metrics can be described mathematically as
(6)$$corr\left(x,y\right)=\frac{\sum_{i=1}^{N}\left(x_{i}-\overset{¯}{x}\right)\left(y_{i}-\overset{¯}{y}\right)}{\sqrt{\sum_{i=1}^{N}{\left(x_{i}-\overset{¯}{x}\right)}^{2}\sum_{i=1}^{N}{\left(y_{i}-\overset{¯}{y}\right)}^{2}}}$$
and
(7)$$SNR\left(x,y\right)=10log_{10}\left(\frac{var\left(x\right)}{MSE\left(x,y\right)}\right)$$
where $var\left(x\right)$ represents the variance of vector x, and $MSE\left(x,y\right)=\frac{1}{n}\sum_{i=1}^{n}{\left(x_{i}-y_{i}\right)}^{2}$ represents the mean squared error between vectors x (ground-truth kinematics) and y (decoded kinematics). These two metrics were computed in each measurement block for each participant, and then the group-level metrics were used to compare. Note that in the case of the correlations, the absolute values of participant-level correlations were used to construct the group-level distribution. Additionally, the group-level chance-level performance was also determined. The kinematic signals were shuffled trial-wise to interrupt the relationship between the features and the kinematic signals. This shuffled pair of signals was then used to train chance-level PLS and UKF decoders. Shuffling was repeated 100 times per model and participant, resulting in 9000 chance-level models (10 participants × 100 repeats × 9 unique models; see Figure 1). These models were then applied to the corresponding participant, adaptation, and measurement block to find the corresponding chance-level correlations. The participant-level chance-level performance was chosen from the 95th percentile of the chance-level distribution, determined from 100 models. Finally, the median value across participants was taken as the group-level chance-level performance. The problem of overfitting the chance level is irrelevant in this case because the corresponding data for each participant were never included in the training data pool (see Figure 1).

#### 2.7.2. Comparing REWPLS and PLSREGRESS

The first step in comparing the decoder would be to assess whether REWPLS, based on the kernel PLS algorithm, and PLSREGRESS, based on the SIMPLS algorithm, would be interchangeable or not. The data from the first session were considered in this comparison. The calibration data from 0% EEG measurement blocks were used to train the decoder for both PLS variations (PLSREGRESS + SR-UKF and REWPLS + SR-UKF) for all participants with the number of PLS fixed at 50 components to guarantee that a similar amount of information was used. The decoder was then applied to the data from all measurement blocks. Additionally, we also varied the type of features utilized by the model: sensor-space and source-space features. The correlations and SNRs were used to assess the interchangeability of the two PLS algorithms. Ideally, we expected REWPLS to perform similarly to PLSREGRESS for both sensor-space and source-space features.

#### 2.7.3. Comparing Simulated Online Experiments

The correlations and SNRs were used to compare results across different transfer learning scenarios, as mentioned previously. The goals for these comparisons were to see the feasibility of applying any of these transfer learning scenarios in a real online experiment. Additionally, we wanted to test whether the sensor-space and sour-space features affected the decoding performance. After simplifying the performance metrics by grouping the measurement blocks and kinematics from 6 kinematics to 3 kinematics (position, velocity, and non-linear kinematics), pairwise Wilcoxon signed-rank tests were employed to compare whether the two distributions were different or not. The *p*-values were adjusted to control the false discovery rate (FDR), according to [44].

#### 2.7.4. Decoding Patterns

The PLS weight matrix can also give us crucial neurophysiological insights. The weight matrix informs us how informative each feature (equivalently, each brain region) is to predict the output kinematics. However, caution has to be taken when interpreting the decoding weight, as it might lead to wrong conclusions regarding the original neural source of interest [45]. To ensure the correct interpretation of the decoding patterns, the PLS weight matrix was taken and transformed as follows:(8)$$A=\frac{1}{g}\Sigma_{\tilde{Y}}W\Sigma_{\tilde{Z}}{}^{-\frac{1}{2}}$$

The PLS weight matrix, W, was multiplied by the covariance matrix of the extended features, $\Sigma_{\tilde{Y}}$, and the matrix square roots of the covariance matrix of the extended hand kinematics, $\Sigma_{\tilde{Z}}$ [9,46]. These matrix multiplications were divided by the model-specific global field power (GFP), g, to reduce the variability across different participants [7]. Note that the covariance matrices shown here are properly normalized covariance matrices, not non-normalized covariance matrices used in the training of REW-PLS. The model-specific GFP was computed by first projecting the corresponding training sensor-space EEG signal data pool onto the source space, randomly selecting a single time point, taking the mean over trials, and finding the standard deviation over voxels. Repeating these processes for 10,000 permutations to determine the standard deviation distribution over voxels and finally take the median of this distribution as the model-specific GFP.

Additionally, to visualize the distribution of all kinds of models in the whole feature space, t-distributed stochastic neighborhood encoding (t-SNE) [47] was used to reduce the dimensionality of the feature space from the decoding pattern from 588 (for source-space feature) to 2 features. The t-SNE reduction was performed for each kinematic separately. To account for different scaling across generalized and individual models, the decoding patterns were further normalized by their maxima before applying t-SNE.

## 3. Results

### 3.1. Comparing REWPLS to PLSREGRESS

First, it is crucial to compare the performance between the PLS models trained with REWPLS and PLSREGRESS. The decoding performance is summarized in Figure 2. The scatter plots show linear trends in all cases, signifying the linear relationship between the PLS models trained with REWPLS and PLSREGRESS. In most cases of correlation, the linear trends of both metrics were consistent with the identity line, indicated by an oblique black line. The exceptions were in the non-linear kinematics, where the linear trend visibly deviated from the identity line and toward PLSREGRESS, suggesting that PLSREGRESS performed better than REWPLS in the case of non-linear kinematics for correlations. For SNRs, the linear trend lines were also consistent with the identity line. However, the mismatch between the linear trend and identity lines in non-linear kinematics was less visible than for correlations. Both sensor-space and source-space models performed similarly in most cases.

### 3.2. Decoding Performance

The overall decoding results from the correlation and SNR are summarized in Figure 3 and Figure 4, respectively. The median correlations, SNRs, and their standard deviations are summarized in Appendix A.

The chance level of the correlation was determined to be around 0.14 for the linear kinematics (position and velocity) and 0.07 for the non-linear kinematics (distance and speed) in all cases. We see that, in most cases of Gen, the median correlation was worse than the chance level or borderline/close to the chance level. In GenC, the median correlation was also worse than the chance level in most cases, except in the 50% EEG block of session 2. In the case of Ind, the median correlation was better than chance in the 50% and 100% EEG blocks. IndC also showed a similar performance to Ind. In all four adaptation strategies, there was a tendency for an increase in the correlation from session 1 to session 2, but it dropped from session 2 to session 3. Within the same session, the correlation increased from 0% EEG to 50% EEG due to the model’s update but then decreased in the 100% EEG blocks.

The decoding performance in terms of SNRs is also visualized in Figure 4. In most cases, the median SNR was around −5 dB for the linear kinematics with generalized strategies (Gen and GenC) and −7 dB for the non-linear kinematics. On the other hand, the individualized strategies indicated median SNRs of around −3 dB for the linear kinematics and −4 dB for the non-linear kinematics (excluding 0% EEG of the first session). The general trends in the SNR were similar, as the SNR improved from session 1 to session 2 but degraded from session 2 to session 3. The generalized strategies indicated broader distributions than the individualized strategies.

### 3.3. Comparing Update Strategies

To aid the comparison between the four update strategies, the performance metrics were combined in two ways. First, the metrics from each measurement block were grouped into the same distribution. Second, the kinematics were grouped into position, velocity, and non-linear kinematics. The summary is shown in Figure 5. Additionally, the *p*-values from pairwise two-tailed Wilcoxon signed-rank tests are summarized in Appendix A for each metric and kinematic group.

Regarding correlations, Gen and GenC performed significantly worse than Ind and IndC (*p* < 0.001 in most cases; see Appendix A). Both individual strategies (Ind and IndC) showed a similar median performance, around 0.14 (in position and velocity) and 0.05 (in non-linear kinematics). Despite the minor differences between median correlations, the *p*-values indicated that sensor-space Ind performed significantly better than sensor-space IndC, source-space Ind, and source-space IndC in the case of position and velocity (Appendix A), but not in non-linear kinematics (Appendix A). Gen showed median correlations of 0.07 (in position and velocity) and 0.03 (in non-linear kinematics). GenC had a performance between the generalized and individual models at around 0.10 (in position and velocity) and 0.05 (in non-linear kinematics).

Regarding SNRs, Gen and GenC performed similarly at around −4.5 dB (in position and velocity) and around −6.2 dB (in non-linear kinematics). Furthermore, Ind and IndC also performed similarly at around −2.8 dB (in position and velocity) and −4 dB (in non-linear kinematics). The statistical tests revealed that Gen and GenC performed significantly worse than Ind and IndC in all cases (Appendix A). The *p*-values also showed that sensor-space Ind performed significantly better than sensor-space IndC, source-space Ind, and source-space IndC, despite small differences.

### 3.4. Decoding Patterns for Generalized Model

The group-level decoding patterns are visualized in Figure 6. Only the patterns from Ind1, Ind2, and Ind3 are shown due to the similarity across the different models. The decoding patterns of every model are additionally visualized in Appendix A.

Despite having the same task in common in all three sessions, the group-level decoder exhibited distinctly different patterns. There were some similarities across the three sessions, for example, the dependency of the information in the paracentral (PCL), both regions of the precuneus (aPCU and pPCU), and cuneus (CU) to explain the velocity. These dependencies shifted from large areas in PCL, PCU, and CU in session 1 and concentrated primarily around PCU in sessions 2 and 3. Overall, the linear kinematics were explained largely by regions in the medial part of the brain, which were the medial pre- and postcentral gyri (mPrCG and mPoCG), both regions of the superior parietal lobes (SPL), and regions between hemispheres (PCL, PCU, and CU). The dependencies of the non-linear kinematics were mostly spread out without a noticeable common pattern between sessions.

To further understand the relationship between the generalized and individual models, the decoding patterns were reduced to 2D via t-SNE for visualization purposes. These results are illustrated in Figure 7. All generalized models can be clustered around GenC1 (green stars; see Figure 7) without a clear distinction between the other different generalized models. On the other hand, the individualized models were spread out and formed a ring around the cluster of all generalized models. Furthermore, for some participants, individualized models from the same participant can be roughly clustered into smaller regions, forming a smaller cluster around the bigger cluster of the generalized models (e.g., see a plot of the distance pattern in Figure 7), which demonstrated the relationship between the generalized and individualized models.

## 4. Discussion

In this study, we assessed the feasibility of utilizing transfer learning to reduce the calibration time for inferring hand trajectories in a PTT study with attempted hand movement. We simulated four online experimental scenarios with different update strategies without acquiring calibration data such that the participant received real feedback in all measurement blocks. All scenarios began with the same starting LOPO model, but the model was adapted differently, either applying the LOPO model throughout the three sessions available without any further updates (Gen), cumulatively updating the LOPO model with individual data from calibration blocks (GenC), or entirely replacing the LOPO model with an individualized model, either cumulatively across sessions (IndC) or discarding the model entirely (Ind). Furthermore, we compared the performance between the features obtained from the sensor space and source space (via source localization) to see which is more suitable for these transfer learning aspects.

We started by comparing the results between models trained via REW-PLS, which utilizes the modified kernel #2 algorithm [27], and PLSREGRESS, which utilizes the SIMPLS algorithm [25], by fixing the number of PLS latent spaces to 50 components. The plot in Figure 2 indicates quite a consistent correlation and SNR between the two variations of PLS models, at least in the linear kinematics. However, the results suggest inferior distance and speed correlations in REW-PLS compared to PLSREGRESS. The distribution of the results from sensor-space and source-space features was mostly consistent, except in the non-linear kinematics for correlations and SNRs. However, we would argue that the independence of the size of the whole training dataset justifies the differences in performance. On top of that, it was also shown that the modified kernel #2 algorithm required on average 6% less CPU time than the SIMPLS algorithm [48].

After justifying the usage of REW-PLS over PLSREGRESS, we trained REW-PLS while adhering to the principle of Occam’s razor. This was implemented by choosing the optimal point (“knee point”) between the number of PLS latent components and the PLS-predicted correlations (only from the linear kinematics, as PLS can only represent the linear relationship). We relied on the correlation rather than the mean square error or, in our case, the normalized mean square error, NMSE, due to the sensitive nature of the MSE (see Appendix A). We expected an increase in the correlation and a decrease in the NMSE as the number of PLS latent components increases (Appendix A). However, there were some cases where the NMSE behaved non-ideally (see Appendix A). In such cases, the NMSE increased instead. By choosing the knee point from the correlation rather than the maximum or minimum point of the correlation of the NMSE, we accounted for these non-ideal cases while keeping the model sufficiently simple. However, we could not find any explanations for the counterintuitive cases where the NMSE increases together with the correlation.

We then illustrated the decoding performance in the correlation and SNR across nine measurement blocks from three sessions (Figure 3 and Figure 4). In most cases, it was also apparent that the LOPO generalized model without any updates (Gen) performed worse than the chance level. The additional individual information in GenC caused slight increases, but they were still worse than the chance level in most cases. Alternatively, by discarding the data from other participants and instead utilizing the individual information, as in the individual approaches (Ind and IndC), the correlation and SNR improved but were still lower than the chance level in the 0% EEG measurement blocks, where the decoder was taken from the previous session and was not updated with the data from the new session. There were no differences between the results obtained by updating the model by discarding the data from the previous session (Ind) and cumulatively updating it (IndC).

The plots were simplified by fusing the nine measurement blocks and grouping the kinematics (Figure 5) to aid pairwise comparisons between the four update strategies. The Ind and IndC performance was significantly better than that of Gen and GenC (Appendix A). This means that the task was too complicated for the model to generalize across participants. The decoding performance of Ind was generally similar to that of IndC within the same type of feature space, showing that there was no benefit of the cumulative model over the non-cumulative one. The improved results from the GenC model were only due to the inclusion of the training data from that respective participant. The differences in terms of median values were similar between sensor-space and source-space features. However, the range of the distribution was slightly broader for sensor-space features, leading to the conclusion that the sensor-space model performed better than the source-space model in position and velocity (see Appendix A). This was counterintuitive, as the source-space models generally utilized a higher number of PLS components than the sensor-space models (see Appendix A). There were observable drops in performance from 100% EEG of the previous session to the 0% EEG measurement block of the next session in both sensor-space and source-space features. This means that the model did not generalize well across participants or across the sessions of the same participant.

Several studies have shown promising decoding correlations for the executed hand movement trajectory [5,8,9,11,14,15,16,46,49,50]. In contrast, only a few studies have focused on the attempted hand movement trajectory [10,11,14,51,52], which is more applicable to the end user but more complicated to decode due to the missing ground truth and lower decoding performance. The average decoding correlations were generally around 0.3–0.5 in executed movement and around 0.2–0.3 in attempted movement. In this study, we obtained an average correlation of around 0.18 in the sensor-space Ind model (see Figure 5), which is similar to the results in [15] but well below the correlations reported in [10], despite using the same dataset as well as decoders, which should be comparable to our sensor-space Ind model. We speculate that this could be due to differences in how the results are represented. For example, it might be that the participant-level correlation was computed by averaging over the absolute trial-level correlations, thereby slightly overestimating the participant-level correlation. Additionally, the lower correlation might be due to the introduction of the LOPO model as a generalized starting model, slightly lowering the overall average correlation.

Another aspect that we explored was how the sensor-space and source-space features would perform given across-participant and across-session generalization problems. To our best knowledge, this is the first study that explored this aspect. Studies that utilized source-space features can be divided into discrete and continuous domains. In the first domain, research has mainly focused on the SMR-based BCI to classify the user intention and reported superior decoding performance compared to sensor-space features [51,52,53,54,55]. Only a handful of studies have focused on continuously inferring the trajectories from EEG from source-space features, which correspond to the continuous domain [11,12,56], and they all reported worse performance or no improvement in source-space compared to sensor-space features. Our feature extraction process was based on computing the average signals from the predefined ROIs, which could be comparable to those used in [11] and [12]. We also obtained no decoding improvements by utilizing source-space features over sensor-space features in transfer learning. Our results disproved our hypothesis that the source-space model would represent common space across measurement units (participants or sessions) due to the additional individual anatomical information during the source-space feature extraction.

To the best of our knowledge, no studies have assessed the learning effect within the continuous decoding of attempted hand trajectories. The first attempt in this regard was a recent study with three sessions from our group [10], showing non-significant changes in participants’ performance over three sessions. We further extended the analysis from [10] to cover the adaptation of the machine (or, more specifically, the algorithm). The decoding performance showed upward trends between sessions 1 and 2. Still, it showed downward trends between sessions 2 and 3, which were also reported in [10], whose authors hypothesized that the reduced performance in session 3 was because of frustration, as the participant’s improvement did not meet their expectation. The decoding patterns in velocity from the source-space model (Figure 6) showed that changes in the pattern became more focused around the PCU and CU from session 1 to session 3, similar to the decoding patterns presented in [10], which might weakly support the learning aspect of the participant, despite no observable improvement in terms of the decoding performance.

Our study contained some limitations that hindered the interpretation of the mutual learning framework. First, three sessions per participant were inadequate to disentangle the long-term learning effect from other short-term across-session effects (e.g., participant’s mood, motivation, and engagement). Second, the end users were not considered in this study, making it difficult to transfer the BCI to the end users. Several studies have assessed the user’s learning effect with more than three training sessions and showed a positive learning effect from mutual learning between the users and the machine [51,57,58,59,60,61]. However, all these studies focused on the SMR-based BCI, which is not comparable to our case.

Despite the evidence so far showing little merit in employing source-space features over sensor-space features to infer hand trajectory continuously, we think it is crucial to investigate why it is the case that source-space features work better than sensor-space features for the discrete classification problem but not for the continuous regression problem. Moreover, we suggested exploring how to utilize the source-space information better, as hand picking ROIs might not be the optimal way to reduce the number of source-space signals. Source-space features might help bridge the gaps between non-invasive and invasive brain signals. Further questions also remain on how to make hand trajectory decoding from the brain more applicable to the end user, so more research must be performed to pursue the answers to these questions.

## 5. Conclusions

One of the main concerns in BCIs is the time required to collect calibration data, which involves acquiring data without real-time feedback. This usually takes considerable time in an experiment. The results show that the task might be too complicated to be generalized across participants and sessions, and sensor-space Ind performed the best among them. Group-level decoding patterns observed over three sessions showed more localized patterns around the precuneus (PCU), which could weakly support the learning aspect of the participants. However, this was not reflected in the performance trends, which increased from session 1 to session 2 but decreased from sessions 2 to 3. Even though the results suggest no benefits of source-space feature decoding, we would like to further investigate the reason why source-space decoding worked in the classification problem but not in the regression problem and how to improve the performance in source-space decoding, as this might play an important role in bridging the non-invasive and invasive domains in the BCI context.

## Figures and Tables

**Figure 1 sensors-23-03593-f001:**
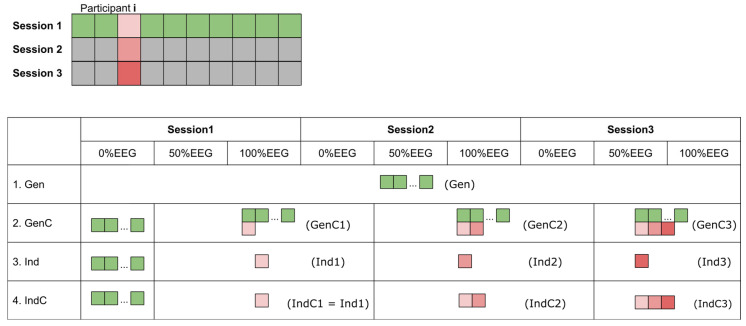
Different update strategies of 4 different use cases: generalized model (Gen), cumulative generalized model (GenC), individual model (Ind), and cumulative individual model (IndC). The blocks represent data included in the training data pool for each measurement block. The green blocks visualize the 0% EEG data from session 1 of every participant except the corresponding participant, represented by the red blocks, where the intensity of the red block represents the corresponding session. The text in parentheses represents the term used to refer to the model with different training data. The suffix C stands for the cumulative model, while the suffix numbers indicate the session where the 0% EEG data are included in the training dataset.

**Figure 2 sensors-23-03593-f002:**
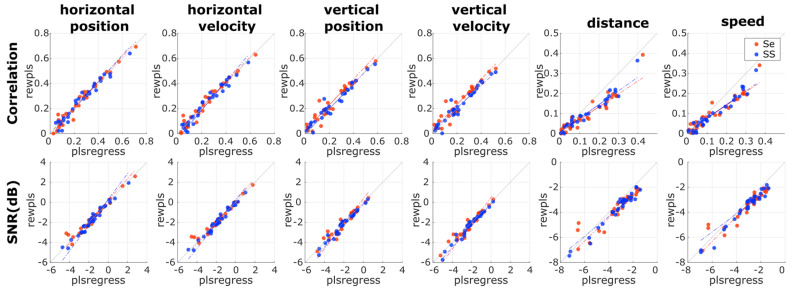
Scatter plots of the participant-level corresponding performance metrics (correlation and SNR) between REWPLS and PLSREGRESS. The metrics from all measurement blocks (0–100% EEG) are plotted in the same scatter plot, resulting in a total of 30 points (10 participants × 3 measurement blocks). The dashed line represents the linear trend of the corresponding points, while the black identity (where x = y) line represents the ideal line, where the performance of REWPLS and PLSREGRESS is the same. The orange and blue colors represent the performance metrics from 2 different feature types: sensor-space (Se) and source-space (SS) features. Note that the distance and speed have different scaling from the other kinematics.

**Figure 3 sensors-23-03593-f003:**
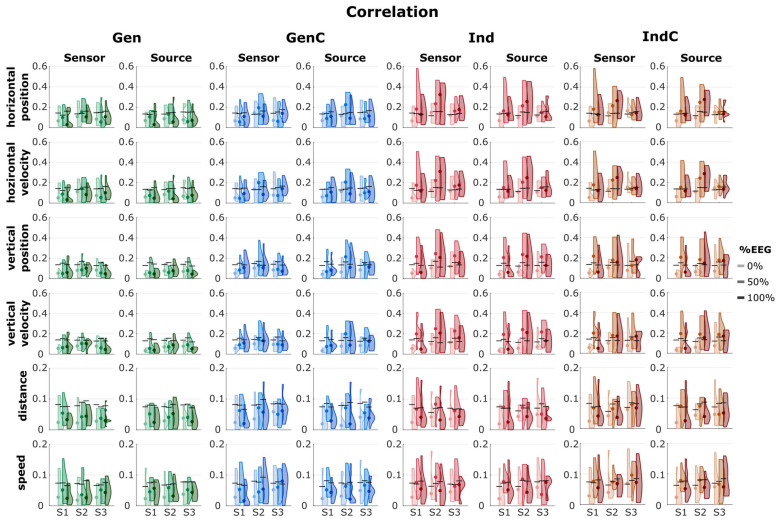
Plots of the distribution of correlations from 4 different scenarios: Gen, GenC, Ind, and IndC, indicated by different colors. The intensity of the color indicates the measurement block, either 0%, 50%, or 100% EEG from sessions 1, 2, and 3, indicated as S1, S2, and S3, respectively. The black horizontal line shows the chance level, while the dot represents the median value of the corresponding measurement block. The corresponding median correlations are summarized in Appendix A.

**Figure 4 sensors-23-03593-f004:**
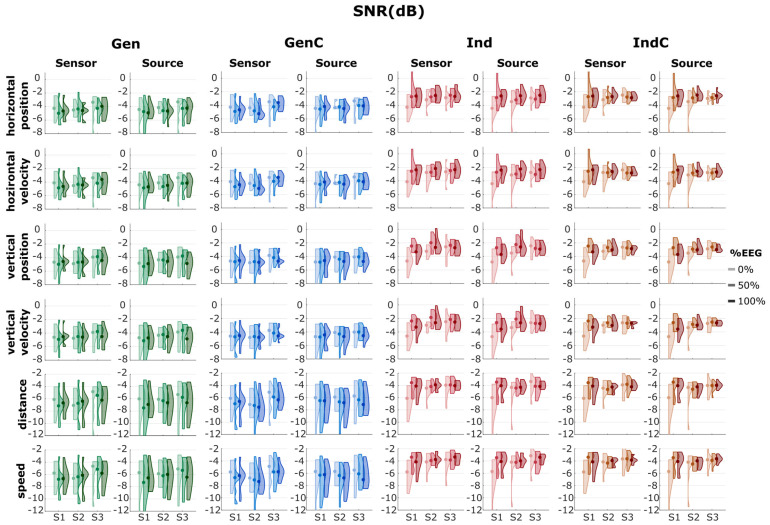
Plots of the distribution of SNRs from 4 different scenarios: Gen, GenC, Ind, and IndC, indicated by different colors. The lightness of the color indicates the measurement block, either 0%, 50%, or 100% EEG from sessions 1, 2, and 3, indicated as S1, S2, and S3, respectively. The corresponding median SNRs and their standard deviations are summarized in Appendix A.

**Figure 5 sensors-23-03593-f005:**
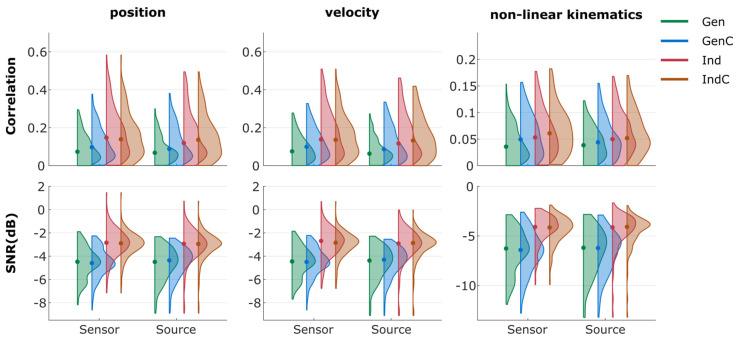
Violin plots of performance metrics. The plots are simplified by merging the measurement blocks and combined into position, velocity, and non-linear kinematics. A dot represents the median value of the corresponding distribution. Color represents the update strategy. Note the different scaling in the non-linear kinematics.

**Figure 6 sensors-23-03593-f006:**
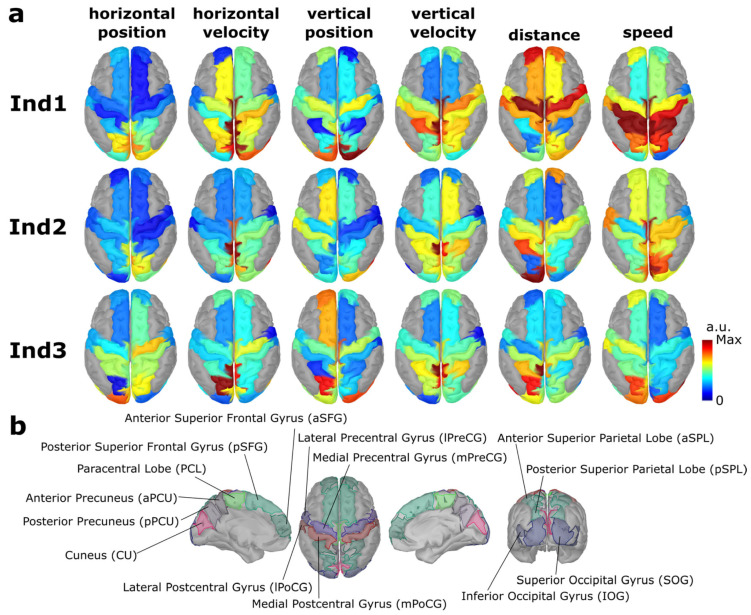
(**a**) Group-level decoding pattern at lag 0 from session 1 (Ind1), session 2 (Ind2), and session 3 (Ind3). Blue and red indicate the value at 0 and maximum. (**b**) The corresponding predefined regions of interest (ROIs). There are 14 ROIs for each hemisphere. The definition of ROIs was based on a Mindboggle atlas [40]. SFG, PCU, PreCG, PoCG, SPL, and OG are further divided into 2 smaller regions with prefixes according to the location (medial/lateral, anterior/posterior, superior/inferior).

**Figure 7 sensors-23-03593-f007:**
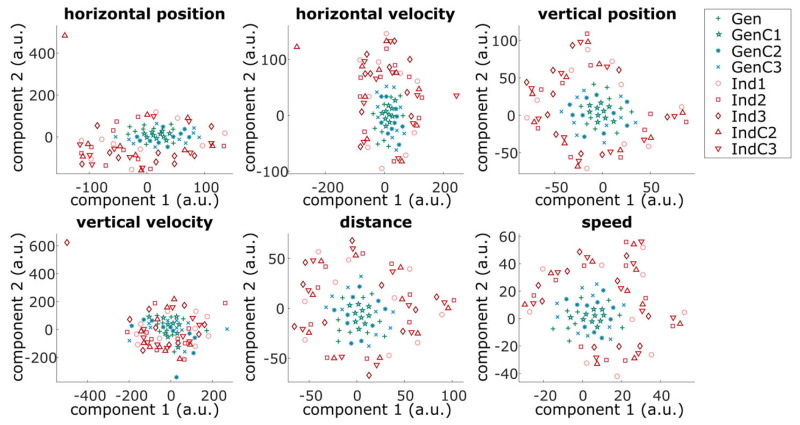
Scatter plots of the reduced dimensionality (via t-SNE [47]) of the decoding patterns of all possible models at different stages of adaptation. Each point represents participant-specific decoding patterns. The different markers represent the different models. The models from generalized and individual approaches are additionally labeled in green and red, respectively.

## Data Availability

Restrictions apply to the availability of these data. Data were obtained from Pulferer et al. [10] and are available from the authors with the permission of Pulferer et al. [10].

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
