# Peer review of "Transfer Learning in Trajectory Decoding: Sensor or Source Space?"

_sensors, 2023, doi:10.3390/s23073593_

Round 1
Reviewer 1 Report
The paper focuses on minimizing the calibration time of the BCI system in the context of continuous hand trajectory decoding with transfer learning. In this paper, the REWPLS method is used to overcome the memory limitation caused by a large amount of training data. The paper builds four models trained with sensor-space features or source-space features and compared their decoding performance. The result showed that sensor-space Ind model indicated better performance than other models. I would require some more improvements. Hereafter I will list revision the Authors should address:
1. In MATERIALS AND METHODS paragraph: “There were 4 measurement blocks in each session: 0% EEG, 50% EEG, 100% EEG, and 100% EEG freerun.”
This sentence should be explained in detail the meaning of the 4 measurement blocks.
2. In MATERIALS AND METHODS paragraph: “where the red blocks (representing the data of the respective participant) were not included in the training pool, and only the green blocks (representing the data from session 1 of every participant) were included.”
This sentence should be more rigorous about the definition of green blocks.
3. In MATERIALS AND METHODS paragraph:
What the three green blocks in the table in Figure 1 represent? Is it the data of the three participants in Session 1 or the data of each participant?
4. In DISCUSSION paragraph: “However, the results suggested inferior distances and speed correlations in REW-PLS compared to PLSREGRESS. The distribution of the results from sensor-space and source-space features was mostly consistent, except in the non-linear kinematics of correlation and SNR.”
A more intuitive data is needed to compare the two methods.
5. In DISCUSSION paragraph:”On top of that, it was also shown that the modified kernel #2 algorithm executed faster than the SIMPLS algorithm.”
This sentence should be introduced with exact numeric values.
Reviewer 2 Report
Authors analyzed across-participant and across-session transfer learning to reduce calibration time of BCI system. The analysis and organization of the approach in the transfering technique for BCI system would be excellent and well described. English grammar looks fine. Therefore, I can recommend this manuscript is minor revision if authors answer the questions as below.
1. Figure 2 label fonts are small and unclear.
2. Figures 3 and 4 size need to be increased a little bit more.
3. Please use abbreviated journal names in reference section.
4. Figure 7 label fonts are not clear to be seen.
5. Please cite the sentence (A typical machine learning model relies on ~) with ref. (https://www.mdpi.com/1424-8220/22/16/6042).
6. Please check font in Lines 128-130 for mathematical expression. I guess autors might change the font for this expression. If not, it is fine.
7. In Line 389, please correct 50%EEG to 50% EEG. Please check others.
8. Authors mentioned that SNR. Why SNR is not that so low such as -3 dB or -4 dB ?
9. How to measure MSE (x,y) as mentioned in Line 293 ?
10. Could authors please describe a little bit more how to obtain the Group-level decoding pattern in Figure 6 ?
Round 2
Reviewer 1 Report
Accept in present form